# Effects of Acupuncture Treatment on Functional Brain Networks of Parkinson's Disease Patients during Treadmill Walking: An fNIRS Study

**Seung Hyun Lee** [1,†][ID]**, Sang-Soo Park** [2,†]**, Jung-hee Jang** [3][ID]**, Sang Hyeon Jin** [4][ID]**, Young-Soo Baik** [5] **and Ho-Ryong Yoo** [6,*]

1 Global Health Technology Research Center, Korea University, Seoul 02841, Korea; aksska82@korea.ac.kr
2 Clinical Trial Center, Dunsan Korean Medicine Hospital, Daejeon 34323, Korea; pssna007@hanmail.net
3 Clinical Medicine Division, Korea Institute of Oriental Medicine, Daejeon 34054, Korea; jee3838@kiom.re.kr
4 Division of Intelligent Robot, DGIST, Daegu 42988, Korea; jinjinsh@dgist.ac.kr
5 Department of Sports Medicine, Konyang University, Chungnam 32992, Korea; bys@konyang.ac.kr
6 Department of Neurology Disorders, Daejeon Korean Medicine Hospital of Daejeon University, Daejeon 35235, Korea
* Correspondence: horyong.yoo@gmail.com; Tel.: +82-42-470-9490
† These authors contributed equally to this work.

**Abstract:** Acupuncture is increasingly being used as an alternative treatment for patients with Parkinson's disease (PD). We aimed to evaluate the effects of acupuncture on gait-related brain function in patients with PD using functional near-infrared spectroscopy (fNIRS). Twenty-four patients with PD were randomly assigned to intervention (acupuncture twice a week for 4 weeks) or control (non-treatment) groups. fNIRS experiments applying a block design were performed at baseline (0 weeks) and 4- and 8-week follow-up and cortical activation and connectivity were evaluated. After acupuncture treatment, oxy-hemoglobin (HbO) levels in the intervention group were significantly increased in the primary motor cortex (M1), supplementary motor area (SMA), and prefrontal cortex (PFC). Furthermore, following acupuncture treatment in the intervention group, the connectivities in the M1 and PFC regions increased. The results show that acupuncture may be a useful complementary treatment for gait disturbances in patients with PD, and fNIRS can be applicable to evaluate neural plasticity directly. The evaluation method in this paper can be used to assess the neural plasticity related to various rehabilitation techniques.

**Keywords:** functional near-infrared spectroscopy; acupuncture; Parkinson's disease; brain connectivity

## 1. Introduction

Parkinson's disease (PD) is one of the most common neurodegenerative disorders. Clinical assessment and diagnosis of PD are based on patient symptoms, and the accuracy of the diagnosis is considered to be approximately 75–95% [1]. However, PD symptoms may vary depending on the patient, and the importance of early diagnosis and treatment of PD is increasing because of the possibility of developing other diseases such as dementia. Signs of PD can be separated into the motor and non-motor symptoms. Among them, motor symptoms mainly include bradykinesia, muscle stiffness, resting tremor, and gait disturbances, and in particular, the prevalence of fractures due to gait disorders is noted to be high [2]. The problems associated with the gait worsen as the disease progresses, which is a significant disease burden that significantly affects the independence and quality of life [3,4].

Many complementary and alternative medicines, including oriental medicines, have recently been used to treat PD [5]. Among them, acupuncture is one of the most active fields of study relevant to PD and has been proven useful in basic and clinical studies [6]. Concerning the effectiveness of acupuncture for treating gait disturbances in PD patients, several clinical studies have also demonstrated its ability to improve gait speed [7,8]. However, systematic research on the mechanism of acupuncture in clinical trials is relatively scarce, and there is also debate about the scientific validity of the effects of acupuncture.

To confirm the mechanism of acupuncture treatment, it is necessary to monitor changes in the cortical region because of neuroplasticity caused by the treatment. Damaged brain function promotes neuroplasticity through repetitive treatment and rehabilitation, such that brain tissue in the cortical region can be reorganized to restore damaged function [9]. Neuroimaging has recently been used as one of the tools to identify the mechanism of acupuncture treatment. Functional magnetic resonance imaging (fMRI) is the most commonly used tool for functional neuroimaging in acupuncture studies because it can indirectly measure brain activity and brain function changes without harmful radiation and invasive procedures. In studies where fMRI was used to measure the effects of acupuncture, each acupoint was found to have a specific effect, and acupuncture was shown to improve symptoms in diseased patients [10,11]. However, fMRI has a limited ability to measure gait-related movements. Moreover, it cannot be used for real-time gait measurements under gravitational acceleration, an essential element of walking. These examples expose one confusing aspect of neuroimaging: it cannot be concluded that fMRI can identify all brain regions involved in behavioral rehabilitation or treatment unless the fMRI procedure is precisely the same as that used in therapy.

Recently, a new technique for brain imaging, functional near-infrared spectroscopy (fNIRS), has gradually gained acceptance and become widely used in clinical practice as a method for diagnosis in psychiatry and neurosurgery [12]. Similar to fMRI, fNIRS is a non-invasive brain imaging technique that can measure blood flow changes in the cerebral cortex. fNIRS can be designed to be relatively insensitive to head motion and uses compact instruments that make it possible to apply to patients with PD at the bedside or in the rehabilitation facility. Several studies have shown, using fNIRS, that oxy-hemoglobin (HbO) levels within the prefrontal cortex are increased in patients with PD during the gait [13,14]. Our previous study also applied fNIRS to PD patients and demonstrated the feasibility of using fNIRS to assess the effects of gait disturbance therapies in PD patients [15]. However, previous studies only showed hemodynamic responses in the cortices following tasks.

In the case of PD, it is important to monitor the connectivity of patients with Parkinson's brain regions and brain activity because the nerve cells that transmit signals are damaged. Moreover, it is important to monitor connectivity because the function can be restored by reorganizing neuroplasticity connectivity as interactions between different regions increase and through the activation of specific regions [16,17]. fNIRS is an important tool for assessing neurophysiological activity during movements such as walking. However, fNIRS has never been used to assess the effects of acupuncture on improving gait disturbances in PD patients. In this study, we identify the effects of acupuncture on brain function in PD patients based on hemodynamic response and connectivity measured by fNIRS during treadmill walking tasks.

## 2. Materials and Methods

### 2.1. Study Design

The study protocol was approved by the Institutional Review Board (IRB) of the Clinical Trial Center at Dunsan Korean Medicine Hospital, Daejeon University (DJDSKH-17-BM-20), as previously published [18]. Patients with PD and gait disturbances who met the inclusion criteria were enrolled and randomly assigned to the control group (conventional treatment alone) or intervention group (acupuncture + conventional treatment). The duration of the study was 8 weeks, which involved the

intervention phase (4 weeks): the subjects received acupuncture two days a week, and the follow-up phase (4 weeks): the subjects were assessed for sustained acupuncture effects. The control group did not receive acupuncture treatment and were followed-up for 8 weeks (Figure 1).

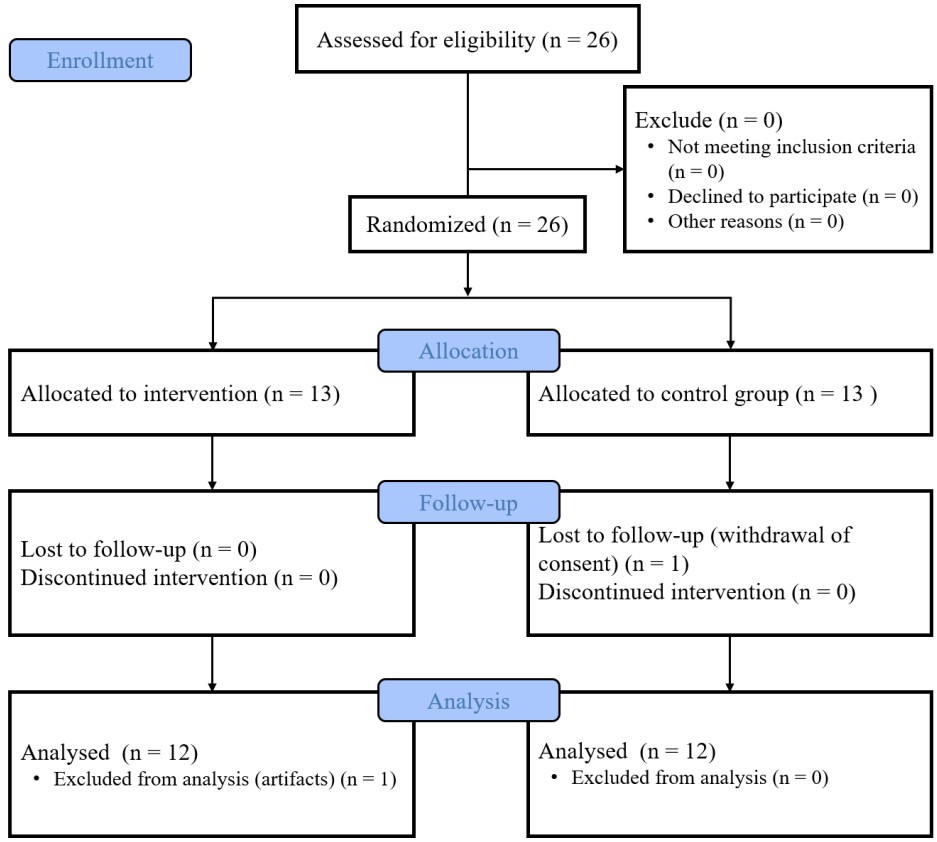

**Figure 1.** Flow chart of the study process.

## 2.2. Participants

Twenty-four patients with PD (15 males, 9 females) participated in this study and were randomly divided into an intervention group (IG) of 12 subjects (9 males, 3 females) and a control group (CG) (*n* = 12: 6 males, 6 females). In this study, a participant in IG and two participants in CG were dropped out. All had significant gait deficits but were able to walk on a treadmill without physical assistance. All participants gave their informed consent for inclusion before they participated in the study. The methods used in this study were performed following the guidelines approved by the IRB. Table 1 shows the baseline characteristics of the participants.

**Table 1.** Baseline characteristics of the participants.

|  | Intervention Group | Control Group | *p*-Value |
|---|---|---|---|
| **Males, *n* (%)** | 10 (76.92%) | 7 (53.85%) | 0.2162 |
| **Females, *n* (%)** | 3 (23.08%) | 6 (46.15%) | |
| **Age (years)** | 65.38 ± 7.81 | 61.46 ± 8.33 | 0.2274 |
| **Age at onset (years)** | 58.46 ± 10.10 | 53.08 ± 9.26 | 0.1693 |
| **Disease duration (years)** | 6.92 ± 4.83 | 8.38 ± 3.88 | 0.3917 |
| **Hoehn & Yahr scale score** | 1.92 ± 0.64 | 1.85 ± 0.69 | 0.7706 |

Values are presented as mean ± SD unless otherwise specified.

### 2.3. Inclusion and Exclusion Criteria

Inclusion criteria consisted of the following: (1) a diagnosis of PD made by a neurologist; (2) ability to walk 10 m; (3) Hoehn an Yahr scale stage 1–4; (4) a stable dose of conventional treatment for at least one month before enrollment [18].

Exclusion criteria consisted of the following: (1) a clinically significant abnormal laboratory value or clinically significant unstable medical or psychiatric illness; (2) any disorder that may interfere with drug absorption, distribution, metabolism, or excretion; (3) the evidence of clinically significant gastrointestinal, cardiovascular, endocrine, or other disorder; (4) treatment for gait disturbance within the last 2 weeks; (5) participation in another clinical trial within the last 4 week; (6) inability to undergo fNIRS [18].

### 2.4. Intervention

A 4-week course of acupuncture treatment was designed to evaluate the effects of acupuncture therapy. Participants in the IG received acupuncture two days a week for 4 weeks. Acupuncture was applied on the dorsal side of the body. The acupoints include the following: Yanglingguan (GB34), Chengfu (BL36), Weizhong (BL40), Chengshan (BL57), Feiyang (BL58), Sanyinjiao (Sp6), Huantiao (GB30), Yaoyan (EX-B7), Fengchi (GB20), Tianzhu (BL9), Shenshu (BL23), Dachangshu (BL25), Guanyuanshu (BL26), and Yongquan (KI1). All acupuncture procedures performed in the IG complied with the STRICTA (Standard for Reporting Intervention in Clinical Trials of Acupuncture) guideline [19], while those in the CG did not receive acupuncture. The CG received conventional treatment alone, including levodopa, catechol-O-methyl transferase inhibitor, monoamine oxidase inhibitor, and dopamine agonists.

### 2.5. Measurement

In the previous study, gait parameters were measured as the primary outcome using the GAITRite system (CIR Systems Inc., Sparta, NJ, USA) for primary outcomes, and MDS-UPDRS (movement disorder society—unified Parkinson's disease rating scale) [20] were partially used to evaluate the gait motor functions [15]. Since this study evaluates the effect of acupuncture on PD's brain network, the analysis of connectivity was conducted based on the fNIRS signals.

fNIRS Measurement

The experiment designed for assessing cortical activation involved the subject walking on a motorized treadmill. The experiment was performed three times (at 0, 4, and 8 weeks) in this study. A relatively low speed (0.1 km/h) was chosen during the task to minimize additional cortical activation due to discomfort and to ensure that patients did not begin to suffer anxiety. The experimental session was arranged in a block paradigm, which lasted 150 s with a post-rest period of 30 s, and was divided into two trials. Each trial consisted of rest and task periods, each with a 30 s duration, with alternate rest and task periods (Figure 2a). All participants performed a treadmill walking assignment in three sessions a day.

A commercial continuous-wave fNIRS system (NIRScout, NIRSx Medical Technology, Berlin, Germany) was used to measure the cortical activation during gaiting on the treadmill. This system performed near-infrared topographic measurements at two different wavelengths (760 and 850 nm) at a sampling rate of 4.17 Hz. The optodes were positioned based on the 10–20 international electrode system. The system consists of 30 optodes (15 transmitters and 15 receivers), which enabled the recording of hemodynamic responses covering the whole brain. The optode configuration resulted in 47 channels, including the primary somatosensory area (SM1), premotor cortex (PMC), supplementary motor area (SMA), and frontal cortex (PFC) (Figure 2b). The midline of channels 11 and 12 (i.e., transmitter 4) was placed in Cz. Optode position registration was conducted using a 3D magnetic digitizer stylus (PATRIOT, Polhemus, Colchester, VT, USA).

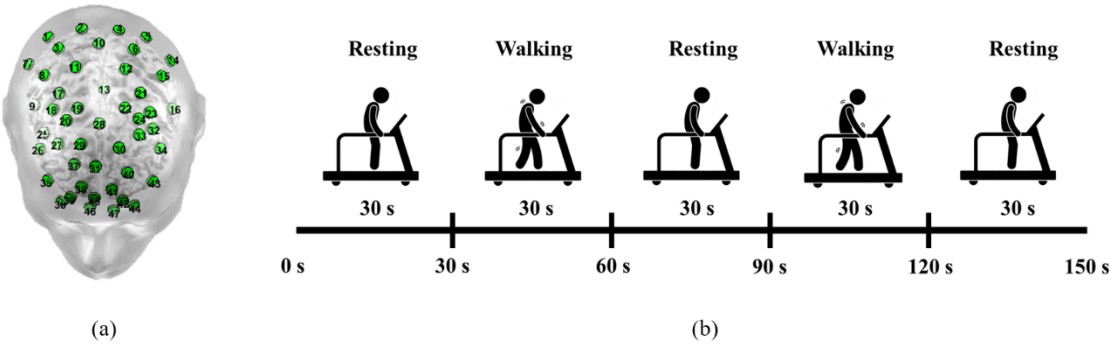

(a)　　　　　　　　　　　　　　　　　　　　　　　(b)

**Figure 2.** Experimental setup and protocol. (**a**) functional near-infrared spectroscopy (fNIRS) channels configuration and (**b**) treadmill walking protocol used in the experiment.

## 2.6. Data Analysis

### 2.6.1. Preprocessing

The analysis was performed using the NIRS-SPM open-source software package implemented in MATLAB (MathWorks, Inc., Natick, MA, USA). The concentration changes in oxy-hemoglobin (HbO) for each channel were obtained using the modified Beer–Lambert law [21] with constant differential pathlength (DPF). The DPFs were set at 7.15 for the wavelength of 760 nm and 5.98 for the wavelength of 850 nm as recommended in the software guideline. Only alterations in HbO were reported as these were shown to be more reliable and sensitive to changes in cerebral blood flow [22–24]. The hemoglobin concentrations were subsequently subjected to independent component analysis (ICA) to eliminate typical motion artifacts [25,26]. Then, a re-referencing method was applied to the NIRS signals to remove common noise [27,28]. The hemodynamic response function (hrf) was applied for the correction of noise from the fNIRS system, and the wavelet-minimum description length (MDL)-based detrending algorithm was used for the correction of global trends due to breathing, heartbeat, or other experimental influences [29]. The block averaging, which is common in the neuroimaging field by repeating the task several times and considering the average per time-points of all repetitions, was also used to remove physiological changes that are not associated with neuronal activity [30,31].

### 2.6.2. Time-Series Analysis

The primary motor cortex (M1), supplementary motor area (SMA), and prefrontal cortex (PFC) were selected as the regions of interest (ROIs). During the task, the concentration changes of HbO were measured from each channel of the ROIs. For each ROIs, the grand average of the hemodynamic response was computed. To compare the activities among the different brain regions, the peak values of the concentration changes of HbO were obtained from the averages of the ROI channels (M1: channels 2, 4, and 10; SMA: channels 11, 12, 13, 17, 19, 21, and 22; PFC: channels 38, 39, 41, 42, 45, 46, and 47). A total of 33 and 39 data samples were used for CG and IG, respectively (CG: 33 samples = 3 trials × 11 subjects, IG: 39 samples = 3 trials × 13 subjects).

### 2.6.3. Statistical Analysis

For the statistical analysis, a two-sample *t*-test and paired sample *t*-test were used to determine the significance of the differences in hemodynamic responses between the two groups (CG vs. IG) and the differences among phases in the ROIs, respectively. Power analysis was performed in G*Power (version 3.1.9.6) [32]. We report the required sample size at 95% power for the study to detect the differences in peak change on HbO concentration in the ROIs. The required sample size was calculated based on mean differences in peak change on HbO between the IG and the CG at the $\alpha$ error = 0.05 and effect size = 0.8. The required minimal total sample size would be 33 samples. Significant changes in HbO were considered to have occurred when $p < 0.05$ [33].

## 2.6.4. Functional Connectivity Analysis

Functional connectivity analysis was performed using the OptoNet software package implemented in MATLAB (MathWorks, Inc., Natick, MA, USA) [24]. OptoNet offers options and choices for the network analysis of specific task blocks or the entire signal length for each subject and trial. OptoNet also provides visual brain connectivity to identify the cortical network, and various estimations were available. In this study, we use the following set of parameters in the OptoNet; signal duration: total length, the number of trials: all trials, the number of subjects: all subjects. The cortical connectivity results were calculated using the coherence between channels and the coherence threshold: 0.9. The coherence is defined as follows [34]:

$$C_{xy}(f) = \frac{|S_{xy}(f)|}{\sqrt{S_{xx}(f)S_{yy}(f)}} \tag{1}$$

## 3. Results

### 3.1. Spatial Registration

The measurement channels were aligned to MNI space using NFRI' fNIRS software [35]. Each measuring channel was associated with the highest percentage of the brain regions. The coordinates of the MNI and anatomical labeling (Brodmann areas) were shown in Table 2.

**Table 2.** The MNI coordinates and associated brain regions were corresponding to all the channels.

| Ch | MNI x | MNI y | MNI z | Region | Ch | MNI x | MNI y | MNI x | Region |
|----|----|----|----|----|----|----|----|----|----|
| 1 | 37 | −34 | 72 | SM1 | 25 | 8 | 31 | 63 | SMA |
| 2 | 17 | −33 | 79 | SM1 | 26 | −11 | 32 | 62 | PMC |
| 3 | −4 | −38 | 78 | M1 | 27 | −32 | 28 | 55 | DLPFC |
| 4 | −21 | −32 | 76 | SM1 | 28 | 39 | 41 | 41 | DLPFC |
| 5 | 49 | −22 | 65 | SM1 | 29 | 25 | 42 | 50 | FEF |
| 6 | 28 | −21 | 75 | M1 | 30 | 12 | 42 | 56 | FEF |
| 7 | 10 | −20 | 79 | PMC | 31 | −3 | 42 | 54 | FEF |
| 8 | −13 | −20 | 79 | PMC | 32 | −18 | 42 | 53 | FEF |
| 9 | −35 | −21 | 73 | PMC | 33 | −34 | 39 | 44 | FEF |
| 10 | 38 | −10 | 69 | SMA | 34 | 44 | 51 | 26 | FPA |
| 11 | 16 | −7 | 76 | SMA | 35 | 20 | 51 | 46 | FEF |
| 12 | −7 | −8 | 77 | SMA | 36 | 6 | 49 | 50 | FEF |
| 13 | −23 | −11 | 75 | SMA | 37 | −13 | 51 | 48 | FEF |
| 14 | 44 | 5 | 60 | PMC | 38 | −38 | 51 | 30 | FPA |
| 15 | 24 | 7 | 71 | PMC | 39 | 14 | 58 | 40 | DLPFC |
| 16 | 10 | 5 | 74 | PMC | 40 | −8 | 58 | 42 | DLPFC |
| 17 | −13 | 6 | 73 | PMC | 41 | 33 | 63 | 20 | FPA |
| 18 | −34 | 5 | 66 | PMC | 42 | 24 | 63 | 29 | FPA |
| 19 | 33 | 19 | 62 | SMA | 43 | 4 | 63 | 32 | FPA |
| 20 | 15 | 19 | 68 | PMC | 44 | −17 | 63 | 30 | FPA |
| 21 | −4 | 18 | 68 | PMC | 45 | −27 | 64 | 20 | FPA |
| 22 | −21 | 21 | 66 | SMA | 46 | 15 | 70 | 22 | FPA |
| 23 | 40 | 28 | 52 | FEF | 47 | −11 | 70 | 20 | FPA |
| 24 | 20 | 32 | 60 | FEF | | | | | |

Ch: Channel, SM1: primary somatosensory area, M1: primary motor cortex, PMC: premotor cortex, SMA: supplementary motor area, FEF: frontal eye fields, DLPFC: dorsolateral prefrontal cortex, FPA: frontopolar area.

### 3.2. Cortical Activation

The grand average hemodynamic changes of HbO in the ROIs during treadmill walking for each phase (0, 4, and 8 weeks) are compared in Figure 3. The colored lines indicate the phases (0 weeks: red; 4 weeks: blue; 8 weeks: green), and the black lines indicate the task start (solid line) and task end (dashed line). In each phase of the CG ROIs, there were no distinct differences in HbO (Figure 3a–c

and Table 3). On the other hand, stronger cortical activations were observed in the IG following acupuncture treatment (Figure 3d–f and Table 3). In the first (0 weeks) and second phase (4 weeks), the difference in HbO activity was not significant, but in the third phase (8 weeks), the increase in HbO in the M1 and PFC regions was high.

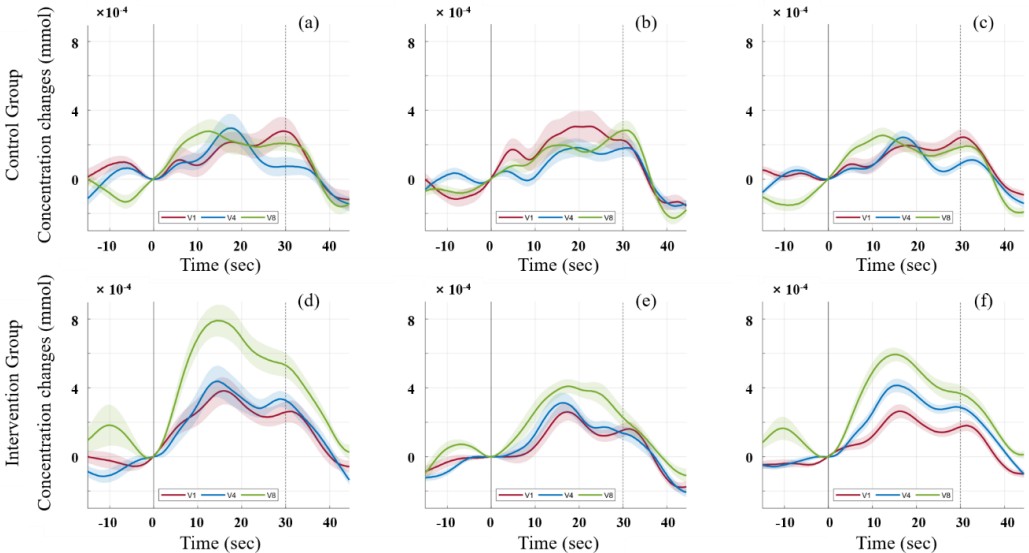

**Figure 3.** The grand average oxy-hemoglobin (HbO) responses for regions of interest (ROIs) in every phase: (**a**) primary motor cortex (M1) in the CG; (**b**) supplementary motor areas (SMA) in the CG; (**c**) prefrontal cortex (PFC) in the CG; (**d**) M1 in the intervention group (IG); (**e**) SMA in the IG; (**f**) PFC in the IG.

To confirm the effectiveness of the acupuncture treatment, the HbO activity of the ROI for each phase (0, 4, and 8 weeks) was compared between the two groups. Figure 4 shows the statistical significance of the difference in HbO peak values between the CG and IG for each phase (0, 4, and 8 weeks). The red bars indicate HbO peak values for the CG, and the blue bars indicate the HbO peak values for the IG. There were some differences between the ROIs in the first phase (0 weeks), but there were no significant differences between the two groups (M1: $p = 0.4542$; SMA: $p = 0.7898$; and PFC: $p = 0.0908$) (Figure 4a and Table 4). Immediately after the acupuncture treatment (4 weeks), the HbO activity was higher in the ROIs of the intervention group, with a significant difference in the SMA (M1: $p = 0.0654$; SMA: $p = 0.0022$; and PFC: $p = 0.1040$) (Figure 4b and Table 4). After 4 weeks of follow-up (8 weeks), both groups had increased HbO activity, with a large increase in the IG. Furthermore, there was a significant difference between the CG and IG in both ROIs (M1: $p = 0.0009$; SMA: $p = 0.000004$; and PFC: $p = 0.0186$) (Figure 4c and Table 4).

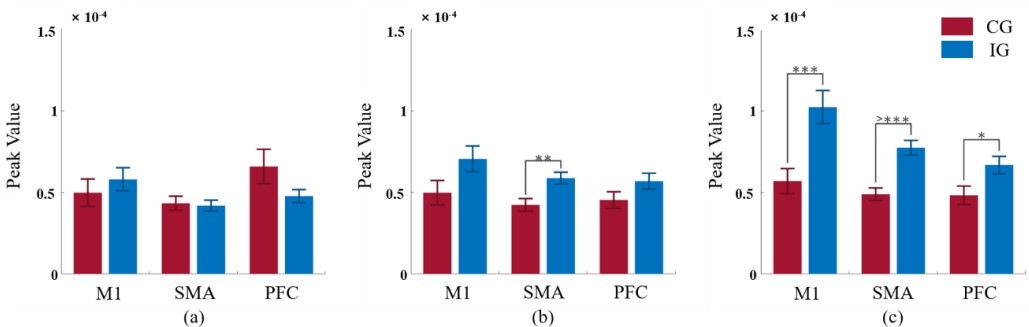

**Figure 4.** Comparison of peak values of group average hemodynamic response between the CG and the IG for each phase: (**a**) the first phase (0 weeks), (**b**) the second phase (4 weeks), (**c**) the third phase (8 weeks). * $p < 0.05$, ** $p < 0.01$, *** $p < 0.001$.

**Table 3.** Average activation in the regions of interest (ROIs) during treadmill walking in patients with Parkinson's disease (PD).

| | | V1 | V4 | V8 | V1 vs. V4 | | V4 vs. V8 | | V1 vs. V8 | |
|---|---|---|---|---|---|---|---|---|---|---|
| | | Mean ± SD (mM) | Mean ± SD (mM) | Mean ± SD (mM) | *t*-Value | *p*-Value | *t*-Value | *p*-Value | *t*-Value | *p*-Value |
| CG | M1 | 0.498 ± 0.00008 | 0.433 ± 0.00004 | 0.658 ± 0.00011 | 0.0005 | 1.00 | −1.224 | 0.23 | −0.827 | 0.415 |
| | SMA | 0.498 ± 0.00007 | 0.423 ± 0.00004 | 0.453 ± 0.00005 | 0.1910 | 0.849 | −1.408 | 0.163 | −1.344 | 0.183 |
| | PFC | 0.569 ± 0.00008 | 0.489 ± 0.00004 | 0.482 ± 0.00006 | 1.9556 | 0.0542 | −0.667 | 0.507 | 1.842 | 0.0694 |
| IG | M1 | 0.58 ± 0.00007 | 0.419 ± 0.00003 | 0.477 ± 0.00004 | −1.708 | 0.0958 | −3.940 | $3.37 \times 10^{-4}$ *** | −5.724 | $1.36 \times 10^{-6}$ *** |
| | SMA | 0.705 ± 0.00008 | 0.587 ± 0.00004 | 0.568 ± 0.00005 | −4.684 | $9.94 \times 10^{-6}$ *** | −3.905 | $1.82 \times 10^{-4}$ *** | −6.836 | $9.44 \times 10^{-10}$ *** |
| | PFC | 1.02 ± 0.0001 | 0.773 ± 0.00005 | 0.667 ± 0.00005 | −1.474 | 0.144 | −1.508 | 0.135 | −3.413 | $9.65 \times 10^{-4}$ *** |

Values are mean ± standard deviation; *** $p < 0.001$.

**Table 4.** Statistical results of peak values in HbO.

| | | CG | IG | *t*-Value | *p*-Value |
| | | Mean ± SD (mM) | Mean ± SD (mM) | | |
|---|---|---|---|---|---|
| **M1** | **V1** | 0.498 ± 0.00008 | 0.58 ± 0.00007 | −0.753 | 0.4542 |
| | **V4** | 0.433 ± 0.00004 | 0.419 ± 0.00003 | 0.267 | $6.54 \times 10^{-2}$ |
| | **V8** | 0.658 ± 0.00011 | 0.477 ± 0.00004 | 1.701 | $9.32 \times 10^{-4}$ *** |
| **SMA** | **V1** | 0.498 ± 0.00008 | 0.705 ± 0.00008 | −1.872 | 0.7898 |
| | **V4** | 0.423 ± 0.00004 | 0.587 ± 0.00004 | −3.105 | $2.20 \times 10^{-3}$ ** |
| | **V8** | 0.453 ± 0.00005 | 0.568 ± 0.00005 | −1.635 | $4.75 \times 10^{-6}$ *** |
| **PFC** | **V1** | 0.569 ± 0.00008 | 1.02 ± 0.0001 | −3.457 | $9.08 \times 10^{-2}$ |
| | **V4** | 0.489 ± 0.00004 | 0.773 ± 0.00005 | −4.731 | 0.104 |
| | **V8** | 0.482 ± 0.00006 | 0.667 ± 0.00005 | −2.376 | $1.86 \times 10^{-2}$ * |

Values are mean ± standard deviation; * $p < 0.05$, ** $p < 0.01$, *** $p < 0.001$.

### 3.3. Functional Connectivity

To provide a detailed account of the brain-wide connectivity of the subjects, we analyzed the functional connectivity of the 47 channels (with a threshold value of 0.9) (Figure 5). Figure 5a–c demonstrates the results of the functional connectivity assessment for each phase of the CG. Functional connectivity tended to decrease and change over time in the CG. Functional connections were connected to a large brain region in the first phase (0 weeks), including the PFC, SMA, and M1 regions in the CG (Figure 5a). Functional connections decreased after 4 weeks compared with the first phase, particularly for the M1 and SMA regions (Figure 5b). Functional connections in the last phase were slightly increased in the M1 region compared to the second phase (Figure 5c). The functional connectivity results of the IG are shown in Figure 5d–f. Functional connectivity was shown to be weak in the M1 and PFC regions associated with gait in the first phase (Figure 5d). After 4 weeks of acupuncture treatment, the functional connectivity decreased in the PFC region; on the other hand, functional connectivity between M1 regions increased (Figure 5e). Figure 5f shows the results of the 4-week follow-up after the treatment with acupuncture. The results show that in contrast to the observations immediately following treatment, the connectivity of the M1 region had decreased, while the connectivity of the PFC region had increased.

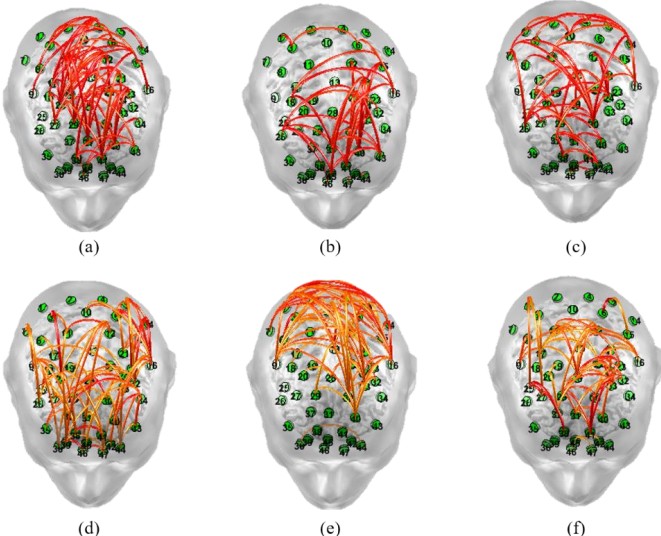

**Figure 5.** Illustrations of functional connectivity network patterns of ROIs before, after a 4-week course of acupuncture treatment, and at 4-week follow-up of PD patients. (**a**) First phase (0 weeks) in the CG, (**b**) second phase (4 weeks) in the CG, (**c**) third phase (8 weeks) in the CG, (**d**) before treatment (0 weeks) in the IG, (**e**) after a 4-week course of acupuncture treatment in the IG, (**f**) after 4-week follow-up in the IG.

## 4. Discussion

fNIRS network analysis was conducted to investigate the mechanism of acupuncture for gait function in patients with PD. Our previous study has confirmed that while there has been no statistically significant change in the neurotransmitter levels in the blood, acupuncture treatment improves gait disturbance in PD patients and may improve balance, cadence, including activation of the cerebral cortex [15]. In this study, we have identified the effects of acupuncture from the perspective of the brain network. The network patterns of task-evoked functional connectivity were analyzed in the whole brain region. By doing this, we tried to validate the hypothesis that acupuncture could modulate the brain network in PD patients. Hemodynamics is dynamically influenced by the formation of network structures over the entire brain. Therefore, the application of a functional connectivity study provides evidence of the mechanism of acupuncture in PD patients.

Our experimental results show an increase in HbO in the M1, SMA, and PFC regions in the IG after 4 weeks of acupuncture therapy. On the other hand, in the CG (without therapy), no significant change in the activity of the ROIs was observed. In addition, the FC pattern was altered in the IG during treadmill walking, including increases in the M1 region and decreases in the PFC region after 4 weeks of acupuncture treatment. Additionally, after 4 weeks of follow-up, FC was increased in the PFC region and decreased comparatively in the M1 region in the IG. On the other hand, in the CG, FC was decreased in the M1 region after 4 weeks and tended to increase slightly after 8 weeks. Based on neuroimaging of human locomotion, two distinct locomotor control pathways have been proposed [36,37]. The direct locomotor pathway is a neural pathway within the central nervous system (CNS) through the basal ganglia, including M1, cerebellum, and spinal cord [37,38]. In contrast, the indirect pathway is a neuronal circuit through the basal ganglia and several associated nuclei within the CNS, including PFC, SMA, and basal ganglia [37,38]. The indirect locomotor pathway is activated when the execution of walking is impaired, and compensatory mechanisms are necessary [13], and these assumptions support the results of our study that the PFC of PD is more activation after acupuncture therapy. The SMA region is known to be involved in gait and balance control [22,39,40], and previous task-related fMRI studies showed decreased activity in the SMA region of patients with PD compared to healthy controls [41]. Therefore, we suggest that this region modulates gait disruption to correct balance in patients with PD. In this study, there was no significant difference in HbO activity between the CG and IG at 0 weeks, but after 4 weeks of acupuncture treatment, there was a significant difference in HbO between the two groups in the SMA region. Furthermore, the increase in HbO in the SMA region in the IG between 0 and 4 weeks proved significant. Our findings show that acupuncture has a mitigating effect on gait dysfunction in PD patients; this effect could be linked to neural mechanisms such as cortical reorganization.

Previous studies have shown that acupuncture can alter the sensory region and change the nerve conduction rate, and these effects were found to be maintained three months after therapy [42,43]. Our results also show that the effect of acupuncture treatment was prolonged, being detected at the 4-week follow-up, which is in line with observations in the literature. In the IG, the HbO of the SMA region was increased 4 weeks after the acupuncture treatment, but there was a significant difference in the SMA and M1 regions at the 4-week follow-up after acupuncture. The peak values of HbO between the two groups, the CG and IG, also differed only in the SMA region at week 4, but there were significant differences in the SMA, M1, and PFC regions at week 8. These effects demonstrate that acupuncture treatment may have therapeutic effects lasting days to weeks.

Functional connectivity refers to the pattern of statistical dependency between various units within the nervous system. The pattern of connectivity is a statistical relationship measured as cross-correlations or coherence. Brain connectivity is, therefore, crucial to understanding how neurons and neural networks process information. Thus, the enhanced functional connectivity in the M1 region resulting from acupuncture, as observed in our study, could also be interpreted as the effects of enhanced motor recovery (Figure 5e). In PD patients, several studies have reported cortical motor reorganization. Frontal cortical activity tends to increase in PD patients during walking and balance

tasks [44], in addition to higher prefrontal activation in PD patients compared to healthy controls during walking and obstacle negotiation, as observed using fNIRS [45]. In our study, increased functional connectivity was observed in the PFC region at follow-up 4-weeks after acupuncture treatment (Figure 5f). These results can also be described in terms of the effects of the recovery of motor and cognitive functions. On the other hand, the CG did not show significant changes in HbO and connectivity in these areas (Figures 4 and 5). These different observations may depend on the effects of acupuncture treatment. Hence, the findings of our current study show that acupuncture could increase the neural plasticity related to motor function by enhancing the M1 and PFC regions in PD patients.

There are some limitations to this study. First, this study was conducted in the assessor-blinded, randomized, controlled, parallel-group trial, and clinical outcomes such as gait parameters and UPDRS (Unified Parkinson's Disease Rating Scale) were also measured. However, Jang et al. studied the primary outcomes and the neurotransmitter levels for acupuncture effects, and the findings were published [15]. In this study, fNIRS data, which are assumed to be related to therapeutic effects, were used. We only considered the brain function to investigate the effectiveness of acupuncture for PD patients. In addition to our proposed evaluation of brain function, it is necessary to analyze with primary outcomes. However, we conducted different tasks to assess gait performance and hemodynamic responses. It is necessary to perform the same task to compare gait performance and hemodynamic responses for future work. A second shortcoming is the use of a treadmill instead of over-ground walking. However, a treadmill was necessary for this study since a stationary fNIRS device was used to measure brain activation. A treadmill velocity of 0.1 km/h was not adjusted due to the risk of falling in participants, and the number of walking trails could not be increased because Parkinson's patients were unable to walk for a long time. Thirdly, it is difficult to judge the effect of each acupoint when several acupoints are used in a session. Further study is needed to clarify the effects of specific acupoints. Finally, the duration of the intervention was short, and the sample size was small. Nonetheless, we have to point out that the findings presented here are just a preliminary study of the effects of acupuncture on the brain network of PD patients. Further studies with a larger sample size and longer duration of intervention are needed to confirm these results.

## 5. Conclusions

In this study, we investigated the effects of acupuncture on brain connectivity. The main findings demonstrate that acupuncture in PD patients mainly induced enhanced functional connectivity and hemodynamic responses in the M1, PFC, and SMA regions compared to the CG patients, which may indicate that the mechanism of acupuncture treatment is related to the promotion of neural plasticity.

**Author Contributions:** Conceptualization, H.-R.Y.; data curation, S.-S.P. and J.-h.J.; formal analysis, S.H.L. and S.H.J.; funding acquisition, H.-R.Y.; methodology, J.-h.J. and Y.-S.B.; supervision, H.-R.Y.; visualization, S.H.L.; writing—original draft, S.H.L. and S.-S.P.; writing—review and editing, S.H.L., S.-S.P., J.-h.J., S.H.J., Y.-S.B., and H.-R.Y. All authors have read and agreed to the published version of the manuscript.

**Funding:** This work was supported by the Korea Health Industry Development Institute (KHIDI) grant funded by the Korea government (grant number: HI15C-0006) and the National Research Foundation of Korea (NRF) grant funded by the Korea government (MSIP) (grant number: NRF-2019R1C1C1011408).

**Conflicts of Interest:** The authors declare that there are no conflicts of interest regarding the publication of this paper.

## Abbreviations

The following abbreviations are used in this manuscript:

| | |
|---|---|
| fNIRS | functional near-infrared spectroscopy |
| PD | Parkinson's disease |
| HbO | oxy-hemoglobin |
| M1 | primary motor cortex |
| SMA | supplementary motor area |
| PFC | prefrontal cortex |
| SM1 | primary somatosensory area |
| PMC | premotor cortex |
| fMRI | functional magnetic resonance imaging |
| IG | intervention group |
| CG | control group |
| IRB | Institutional Review Board |
| ICA | independent component analysis |
| HRF | hemodynamic response function |
| MDL | minimum description length |
| ROIs | regions of interest |
| MDS-UPDRS | movement disorder society—unified Parkinson's Disease Rating Scale |

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
