# Peer review of "Effects of Acupuncture Treatment on Functional Brain Networks of Parkinson’s Disease Patients during Treadmill Walking: An fNIRS Study"

_applsci, doi:10.3390/app10248954_

Round 1

Reviewer 1 Report

The authors compared functional near-infrared spectroscopy (fNIRS) data between Parkinson’s disease (PD) patients with acupuncture treatment and those without it. They found that acupuncture in PD patients mainly induced enhanced functional connectivity and hemodynamic responses in the primary motor cortex (M1), prefrontal cortex (PFC) and supplementary motor area (SMA) regions compared to the control group patients. The results are worth to be published and the writing is clear. The following minor comments are for making this manuscript more comprehensive.

Minor

  1. Is there any cure effect on the symptoms reflecting Hoehn & Yahr scale score after the acupuncture treatment? Please analyze the correlation between the improvement of “Hoehn & Yahr scale score” and “fNIRS data.”
  2. In Figure 5, there is a big difference in the functional connectivity network patterns between “(a) First phase (0 weeks) in the CG” and “(d) before treatment (0 weeks) in the IG.” Why they are so different? Please explain the reason.
  3. Were there any differences in the hemodynamic responses and the functional connectivity network patterns between males and females? Please show the compared data between males and females.
  4. There is no description about the acupoint they used. The authors described that it is difficult to judge the effect of each acupoint when several acupoints are used in a session. It is a big limitation in this study. If they recorded the details of the acupoints, it would be possible to stratify the data in each acupoint and analyze the correlation between the acupoints and the fNIRS data. Anyway, please describe acupoints and the treatment method as much as they can.
  5. Please discuss about the differences in fNIRS results between the PD patients treated with the acupuncture in this study and those treated with levodopa in other papers.
  6. There is a typo in line 100. Please change “TG” to “IG.”

Author Response

[Dec. 10, 2020]

Takayoshi Kobayashi, Ph.D

Editor-in-Chief

Applied Sciences

Original Manuscript ID: Applsci-1024978

Dear Editor,

We would like to thank you and the reviewers for your comments on our manuscript “Effects of Acupuncture Treatment on Functional Brain Networks of Parkinson’s Disease Patients during Treadmill Walking: An fNIRS Study” We have revised the manuscript to address the reviewers’ concerns and attached a point-by-point response to each of the reviewers’ comments below.

We greatly appreciate the opportunity to have our manuscript reviewed and considered for publication in Applied Sciences. We hope that the revised manuscript is now more appropriate to the readership of your esteemed journal.

We look forward to hearing from you.

Yours sincerely,

Horyong Yoo

Clinical Trial Center, Dunsan Korean Medicine Hospital

Daejeon University

75 Daedeok-daero 176 Beon-gil, Seo-gu, Daejeon 35235

Republic of Korea

Tel: +82-42-470-9490

Fax: +82-42-470-9486

Reviewer#1, Concern # 1:

Is there any cure effect on the symptoms reflecting Hoehn & Yahr scale score after the acupuncture treatment? Please analyze the correlation between the improvement of “Hoehn & Yahr scale score” and “fNIRS data.”

Author response: 

Our previous study showed no significant differences between the groups in “Hoehn & Yahr scale” [1]. Because the intervention duration was short (only 4 weeks), it’s hard to identify the improvement of “Hoehn & Yahr scale”. The assessments of gait performance and hemodynamic response were conducted during different overground and treadmill walking tasks, respectively. Therefore, it was difficult to confirm a direct correlation between gait improvement and hemodynamic changes in this study.

However, we investigated the correlations between gait performance and hemodynamic change. Correlation analysis was performed on the gait parameters (cadence, stride time, swing time, and single support time) and the oxy-Hb level of the region of interest (ch 11, 15, 35, 39, and 40) that were found to be statistically significant between the intervention and control groups. Additionally, each parameter had a delta value (week 4 – week 0 or week 8 – week 0). The delta (week 8 – week 0) activities of the PFC (ch 35 and 39) were positively correlated with the delta (week 8 – week 0) swing time and single support time in the intervention group (r = 0.640, p = 0.019 and r = 0.652, p = 0.016, respectively).

We added a limitation on this to the Discussion section (lines 333 ~ 336)

In addition to our proposed evaluation of brain function, it is necessary to analyze with primary outcomes. However, we conducted different tasks to assess the gait performance and hemodynamic responses. It is necessary to perform the same task to compare gait performance and hemodynamic responses for future work.

Reviewer#1, Concern # 2:

In Figure 5, there is a big difference in the functional connectivity network patterns between “(a) First phase (0 weeks) in the CG” and “(d) before treatment (0 weeks) in the IG.” Why they are so different? Please explain the reason.

Author response: 

Thank you for your comments. We showed no statistically significant difference between the control and intervention groups in brain activation (Table 4). In the connectivity analysis, the coherence threshold was set to 0.95 in this paper. It indicates a strong positive linear relationship. So, there was only few connection in the intervention group. We modified the coherence threshold to 0.9 and performed the network analysis again.

We have made the appropriate corrections in the Materials and Methods section and modified Figure 5 in the Results section of the revised manuscript

2.6.4 Functional Connectivity Analysis (lines 199)

The cortical connectivity results were calculated using the coherence between channels and the coherence threshold: 0.9.

Figure 5. Illustrations of functional connectivity network patterns of ROIs before, after a 4-week course of acupuncture treatment, and at 4-week follow-up of PD patients. (a) First phase (0 weeks) in the CG, (b) second phase (4 weeks) in the CG, (c) third phase (8 weeks) in the CG, (d) before treatment (0 weeks) in the IG, (e) after a 4-week course of acupuncture treatment in the IG, (f) after 4-week follow-up in the IG.

Reviewer#1, Concern # 3:

Were there any differences in the hemodynamic responses and the functional connectivity network patterns between males and females? Please show the compared data between males and females.

Author response:

Thank you for your thoughtful comments. In line with your precious comments, we compared network data between males and females in two groups. In the intervention group, the connectivity between females and males was different, and the connectivity of the males was similar to the results of the group analysis (Figure 5). However, there was no difference between females and males in the control group. In the present study, the female-male ratio was about 7:3 in the intervention group, and in the control group, it was about 5:5 (Table 1). So, it is thought that it will be difficult to generalize due to gender imbalance.  It is very difficult to recruit female and male patients in a balance to study. Therefore, further studies are needed to clarify the effects of the gender difference.

Reviewer#1, Concern # 4:

There is no description about the acupoint they used. The authors described that it is difficult to judge the effect of each acupoint when several acupoints are used in a session. It is a big limitation in this study. If they recorded the details of the acupoints, it would be possible to stratify the data in each acupoint and analyze the correlation between the acupoints and the fNIRS data. Anyway, please describe acupoints and the treatment method as much as they can.

Author response: 

We agree with your comment. We have added the explanations on acupoints in the Materials and Methods sections of the manuscript, and we also described a limitation on the Discussion section.

2.4 Intervention (lines 120 ~ 123)

The acupoints include the following: Yanglingguan (GB34), Chengfu (BL36), Weizhong (BL40), Chengshan (BL57), Feiyang (BL58), Sanyinjiao (Sp6), Huantiao (GB30), Yaoyan (EX-B7), Fengchi (GB20), Tianzhu (BL9), Shenshu (BL23), Dachangshu (BL25), Guanyuanshu (BL26), and Yongquan (KI1).

Discussion (line 341 ~ 342)

Thirdly, it is difficult to judge the effect of each acupoint when several acupoints are used in a session. Further study is needed to clarify the effects of specific acupoints.

Reviewer#1, Concern # 5:

Please discuss about the differences in fNIRS results between the PD patients treated with the acupuncture in this study and those treated with levodopa in other papers.

Author response:  

In this study, the control group (CG) received conventional treatment without acupuncture treatment. We have made the appropriate corrections in the Materials and Methods section of the revised manuscript as follows:

2.1 Study Design (lines 89 ~ 91)

Patients with PD and gait disturbances who met the inclusion criteria were enrolled and randomly assigned to the control group (conventional treatment alone) or intervention group (acupuncture + conventional treatment).

2.4 Intervention (lines 125 ~ 127)

The CG received conventional treatment alone, including levodopa, catechol-O-methyl transferase inhibitor, monoamine oxidase inhibitor, and dopamine agonists.

Reviewer#1, Concern # 6:

There is a typo in line 100. Please change “TG” to “IG.”

Author response: 

(lines 100)

We updated the manuscript by changing “TG” to “IG” in the Materials and Methods section

References

[1]  Jang, J.-H.; Park, S.; An, J.; Choi, J.-d.; Seol, I.c.; Park, G.; Lee, S.H.; Moon, Y.; Kang, W.; Jung, E.-S., et al. Gait Disturbance Improvement and Cerebral Cortex Rearrangement by Acupuncture in Parkinson’s Disease: A Pilot Assessor-Blinded, Randomized, Controlled, Parallel-Group Trial. Neurorehabilitation and Neural Repair 0, 1545968320969942, doi:10.1177/1545968320969942.

Reviewer 2 Report

Interesting work. I have some concerns for the authors:

1) Why did the control group not receive treatment? Are there no alternative therapies for acupuncture?

2) Patients' inclusion and exclusion criteria are undefined.

3) why did the authors targeted the dorsal side of the body?

4) the authors should better argument gait speed selection during treadmill testing. It is indeed known that gait velocity can critically affect brain activation while walking (e.g., Calabrò et al. J Neuroeng Rehabil. 2019)

5) even though this study evaluated the effect of acupuncture on PD's brain networks, it would be interesting to seek whether correlations between clinical and fNIRS signal improvement exists. this would make the ms also interesting in a rehabilitative perspective.

Author Response

[Dec. 10, 2020]

Takayoshi Kobayashi, Ph.D

Editor-in-Chief

Applied Sciences

Original Manuscript ID: Applsci-1024978

Dear Editor,

We would like to thank you and the reviewers for your comments on our manuscript “Effects of Acupuncture Treatment on Functional Brain Networks of Parkinson’s Disease Patients during Treadmill Walking: An fNIRS Study” We have revised the manuscript to address the reviewers’ concerns and attached a point-by-point response to each of the reviewers’ comments below.

We greatly appreciate the opportunity to have our manuscript reviewed and considered for publication in Applied Sciences. We hope that the revised manuscript is now more appropriate to the readership of your esteemed journal.

We look forward to hearing from you.

Yours sincerely,

Horyong Yoo

Clinical Trial Center, Dunsan Korean Medicine Hospital

Daejeon University

75 Daedeok-daero 176 Beon-gil, Seo-gu, Daejeon 35235

Republic of Korea

Tel: +82-42-470-9490

Fax: +82-42-470-9486

Reviewer#2, Concern # 1:

Why did the control group not receive treatment? Are there no alternative therapies for acupuncture?

Author response:

In this study, the control group (CG) received conventional treatment without acupuncture treatment. We have made the appropriate corrections in the Materials and Methods section of the revised manuscript as follows:

2.1 Study Design (lines 89 ~ 91)

Patients with PD and gait disturbances who met the inclusion criteria were enrolled and randomly assigned to the control group (conventional treatment alone) or intervention group (acupuncture + conventional treatment).

2.4 Intervention (lines 125 ~ 127)

The CG received conventional treatment alone, including levodopa, catechol-O-methyl transferase inhibitor, monoamine oxidase inhibitor, and dopamine agonists.

Reviewer#2, Concern # 2:

Patients' inclusion and exclusion criteria are undefined.

Author response: 

Thank you for your thoughtful review. We updated the manuscript by adding the inclusion and exclusion criteria in the Materials and Methods section.

2.3 Inclusion and exclusion criteria (lines 107 ~ 116)

Inclusion criteria consisted of the following: (1) a diagnosis of PD made by a neurologist; (2) ability to walk 10 m; (3) Hoehn an Yahr scale stage 1-4; (4) and a stable dose of conventional treatment for at least one month before enrollment.

Exclusion criteria consisted of the following: (1) a clinically significant abnormal laboratory value or clinically significant unstable medical or psychiatric illness; (2) any disorder that may interfere with drug absorption, distribution, metabolism, or excretion; (3) the evidence of clinically significant gastrointestinal, cardiovascular, endocrine, or other disorder; (4) treatment for gait disturbance within the last 2 weeks; (5) participation in another clinical trial within the last 4 week; (6) inability to undergo fNIRS.

Reviewer#2, Concern # 3:

why did the authors targeted the dorsal side of the body?

Author response: 

Gait ability is a significant goal in rehabilitation exercise because gait is an important ability in achieving functional independence. In this study, the gait-acupuncture method, which was used for patients with gait impairment, was used [2]. This method is an acupuncture method that takes acupoints on the back of the human body (dorsal side). According to the gait-acupuncture method, acupoints were defined on the dorsal side.

We have added the explanations on acupoints in the Materials and Methods sections of the manuscript, and we also described a limitation on the Discussion section.

2.4 Intervention (lines 120 ~ 123)

The acupoints include the following: Yanglingguan (GB34), Chengfu (BL36), Weizhong (BL40), Chengshan (BL57), Feiyang (BL58), Sanyinjiao (Sp6), Huantiao (GB30), Yaoyan (EX-B7), Fengchi (GB20), Tianzhu (BL9), Shenshu (BL23), Dachangshu (BL25), Guanyuanshu (BL26), and Yongquan (KI1).

Discussion (line 341 ~ 342)

Thirdly, it is difficult to judge the effect of each acupoint when several acupoints are used in a session. Further study is needed to clarify the effects of specific acupoints.

Reviewer#2, Concern # 4:

the authors should better argument gait speed selection during treadmill testing. It is indeed known that gait velocity can critically affect brain activation while walking (e.g., Calabrò et al. J Neuroeng Rehabil. 2019)

Author response: 

Thank you for your thoughtful review. 

We intended for the conditions to be identical with overground walking, but it was impossible to walk overground during the fNIRS assessment because our fNIRS systems are not portable. However, they were not allowed to use walking aids to walk on the treadmill, similar to ground walking.

There are some locomotion disturbances in PD characterized by walking in a sped-up fashion with a high cadence, gait destination, and a forward center of gravity while the trunk leans forward involuntarily [3]. The PD patients can walk independently, but the treadmill speed was kept constant for the safety of the PD patients.

We have addressed the explanation about the limitation of treadmill walking in the Discussion section of the manuscript.

(lines 334 ~ 341)

In addition to our proposed evaluation of brain function, it is necessary to analyze with primary outcomes. However, we conducted different tasks to assess the gait performance and hemodynamic responses. It is necessary to perform the same task to compare gait performance and hemodynamic responses for future work. A second shortcoming is the use of a treadmill instead of over-ground walking. However, a treadmill was necessary in this study since a stationary fNIRS device was used to measure brain activation. Also, a treadmill velocity of 0.1 km/h was not adjusted due to the risk of falling in participants, and the number of walking trails could not be increased because Parkinson’s patients were unable to walk for a long time.

Reviewer#2, Concern # 5:

even though this study evaluated the effect of acupuncture on PD's brain networks, it would be interesting to seek whether correlations between clinical and fNIRS signal improvement exists. this would make the ms also interesting in a rehabilitative perspective.

Author response: 

Thank you for your thoughtful review. Our previous study showed no significant differences between the groups in primary outcomes [1]. Because the intervention duration was short (only 4 weeks), it’s hard to identify the improvement of gait parameters. The assessments of gait performance and hemodynamic response were conducted during different tasks on overground and treadmill walking, respectively. Therefore, it was difficult to confirm a direct correlation between gait improvement and hemodynamic changes in this study.

However, we investigated the correlations between gait performance and hemodynamic change. Correlation analysis was performed on the gait parameters (cadence, stride time, swing time, and single support time) and the oxy-Hb level of the region of interest (ch 11, 15, 35, 39, and 40) that were found to be statistically significant between the intervention and control groups. Additionally, each parameter had a delta value (week 4 – week 0 or week 8 – week 0). The delta (week 8 – week 0) activities of the PFC (ch 35 and 39) were positively correlated with the delta (week 8 – week 0) swing time and single support time in the intervention group (r = 0.640, p = 0.019 and r = 0.652, p = 0.016, respectively).

We added a limitation on this to the Discussion section

(lines 334 ~ 337)

In addition to our proposed evaluation of brain function, it is necessary to analyze with primary outcomes. However, we conducted different tasks to assess the gait performance and hemodynamic responses. It is necessary to perform the same task to compare gait performance and hemodynamic responses for future work.

References

[1]  Jang, J.-H.; Park, S.; An, J.; Choi, J.-d.; Seol, I.c.; Park, G.; Lee, S.H.; Moon, Y.; Kang, W.; Jung, E.-S., et al. Gait Disturbance Improvement and Cerebral Cortex Rearrangement by Acupuncture in Parkinson’s Disease: A Pilot Assessor-Blinded, Randomized, Controlled, Parallel-Group Trial. Neurorehabilitation and Neural Repair 0, 1545968320969942, doi:10.1177/1545968320969942.

[2] Lee, J.H, Lee, I.W., Moon, S.H., Kang, J.S., Lim, S.M., An, J.J.,et al. The Effects of Juheng Acupuncture Treatment on Gait Disturbed Patient Caused by Stroke. Korean J Orient Int Med 2007, 77-87.

[3] Delval, A., Rambour, M., Tard, C., Dujardin, K., Devos, D., Bleuse, S., el al. Freezing/festination during motor tasks in early‐stage Parkinson's disease: A prospective study. Movement Disorders 2016, 31(12), 1837-1845.

Round 2

Reviewer 2 Report

the authors addressed all my concerns.